# Gender Differences in the Indirect Effect of Psychosocial Work Environment in the Association of Precarious Employment and Chronic Stress: A Cross-Sectional Mediation Analysis

**DOI:** 10.3390/ijerph192316073

**Published:** 2022-12-01

**Authors:** Fabrizio Méndez-Rivero, Óscar J. Pozo, Mireia Julià

**Affiliations:** 1Research Group on Health Inequalities, Environment, and Employment Conditions (GREDS-EMCONET), Department of Political and Social Sciences, Universitat Pompeu Fabra, 08005 Barcelona, Spain; 2Applied Metabolomics Research Group, IMIM (Hospital del Mar Medical Research Institute), 08003 Barcelona, Spain; 3ESIMar (Mar Nursing School), Parc de Salut Mar, Universitat Pompeu Fabra-Affiliated, 08003 Barcelona, Spain; 4SDHEd (Social Determinants and Health Education Research Group), IMIM (Hospital del Mar Medical Research Institute), 08003 Barcelona, Spain

**Keywords:** cortisol, stress, precarious employment, psychosocial risk factors, gender

## Abstract

Gender differences in the association between precarious employment and chronic stress have been found but the mechanisms underlying this relationship have not been explored. The main objective of this study was to evaluate the mediating effects of psychosocial risk factors at work (i.e., demands, control, and support) and work–life conflicts in the relationship between precarious employment and chronic stress as measured through the production of steroid hormones (both adrenal and gonadal) for men and women separately. Cross-sectional data were derived from a sample of workers from Barcelona (n = 125–255 men; 130 women). A set of 23 markers were determined from hair samples to evaluate the production of both adrenal and gonadal steroids. Decomposition analyses were applied to estimate the indirect effects of psychosocial risk factors and work–life conflict using linear regression models. Gender differences in the association between precarious employment and steroids production were confirmed. Psychosocial risk factors and work–life conflicts had indirect effects only among women (β_Cortisol_ = 0.18; 95% CI: 0.04–0.32; β_Cortisol/Cortisone_ 0.19; 95% CI: 0.08–0.31; β_%Cortisol_ 0.12; 95% CI: 0.05–0.20). Gender differences suggest that the physiological response to precarious employment could be determined by the social construction of gender identities, as well as by positions and roles in the labour market and family. Future studies should delve further into these differences to improve employment and working policies, thus mitigating gender inequalities in the labour market to prevent work-related stress.

## 1. Introduction

Precarious employment (PE) refers to a generalized phenomenon of employment insecurity, income inadequacy, and lack of rights and protection that has become widely extended in recent decades in Europe [1,2,3]. It is recognized as a significant social determinant of workers health, both physical and mental [4]. The Precarious Employment and Stress: The Biomedical Embodiment of Social Factors (PRESSED) project, which includes this study, aims to explain the links between PE and stress [5,6,7].

Stress is associated with poor mental and physical health [8] and increases the risk of suffering from various health problems, such as cardiovascular disease, metabolic syndrome, osteoporosis and/or depression [9], coronary heart disease [10], mental health [11], hypertension [12], and musculoskeletal disorders [13]. For these reasons, stress is a major public health problem in today’s societies. In social epidemiology research, stress is mainly measured through self-reported indicators; however, it is often pointed out that these measures can be affected by biases in the subjective interpretation of stressful events. Furthermore, from the embodiment perspective, the psychosocial work environment can be biologically incorporated, transforming the individual’s physiological characteristics over time while bypassing their consciousness [14]. Such assumptions imply that the embodiment of PE should be measured through biomarkers and not only self-reported indicators. The PRESSED project aims to overcome these limitations by using indicators of steroid hormone production, such as cortisol and its metabolites.

Cortisol is a glucocorticoid steroid hormone involved in the body’s response to stress and, therefore, it is widely used as an indicator of stress [15]. Other steroid biomarkers related to both the hypothalamic–pituitary–adrenhal (HPA) axis (e.g., cortisol metabolites) and the hypothalamic–pituitary–gonadal (HPG) axis, (e.g., testosterone or dehydroepiandrosterone) have recently been used [16,17]. Hair is normally preferred for chronic measurements since it has been shown to be a more accurate and robust medium than serum, saliva, or urine, which may be affected by factors such as circadian rhythm or needle apprehension [18]. In the PRESSED project, cortisol and metabolites from both the HPA and HPG axes were measured in hair to obtain chronic information from both axes. 

Previous studies found associations between stress, temporariness [19], and perceived insecurity [20], two unidimensional measures of PE widely used in the literature. However, unidimensional measures have not been proven to be able to capture the phenomenon of PE in its full extent, and often the results of their association with health have been contradictory [1]. For this reason, the PRESSED project uses a multidimensional measure—the EPRES scale—that has been highlighted as an insightful tool for operationalizing PE, employing an instrument that encompasses six dimensions: temporariness, disempowerment, vulnerability, wages, rights, and the exercise of rights [21].

A recent PRESSED-related study showed an association between PE and both subjective (PSS) and objective (adrenal and gonadal steroid production) chronic stress, although the latter showed significant differences between men and women [6]. The next step within this project is to study the pathways underlying this association that could explain the differences between men and women in physiological responses to PE. 

Some conceptual frameworks suggest that PE may affect health through the psychosocial work environment shaped by the work organisation. On the one hand, from a social epidemiology perspective, the health risks associated with the psychosocial environment are related to the deterioration of working conditions because of the precariousness of employment conditions [4]. On the other hand, from a psychosocial epidemiology approach, it has been pointed out that social processes at the macro-level (e.g., employment conditions) and meso-level (e.g., work organisation) lead to perceptions and psychological processes at the individual level, such as work-related stress. Chronic stress could be produced by the psychosocial work environment through direct psychobiological processes or by modifying behaviours and lifestyles [22,23,24]. Based on this approach, the PRESSED conceptual framework suggests that the psychosocial work environment is an intermediate step in a causal pathway that links economic, social, and political structures with chronic stress through psychological and physiological processes [5]. 

PE conditions have a negative impact on work organization, creating psychosocial environments that are hazardous for the health and well-being of workers. Recent studies suggest that workers in PE are more likely to experience psychosocial risks, such as workplace violence, sexual harassment, bullying, discrimination [25], high-strain jobs [26], and effort–reward imbalance [27,28].

Therefore, recent evidence shows a moderate association between stress-related mental disorders and effort–reward imbalance, high job demands, organizational justice, social support, high emotional demands, and decision-making authority [29]. Strong evidence shows that high job demands, low job control, low support from co-workers, low supervisor support, low procedural fairness, low relational justice, and high effort–reward imbalance can be used to predict the incidence of stress-related disorders [30]. 

According to the International Labor Organization’s definition, psychosocial risk factors extend to extra-organizational aspects, such as domestic demands, which may affect workers’ health [31]. Specifically, work–life conflict (WLC) is a relevant psychological stressor in contemporary working life that has increased amongst employees in most economic sectors [32,33]. It is a form of inter-role conflict whereby fulfilling role demands emanating from the work domain interferes with fulfilling role demands in the home domain or in leisure activities [34]. Strong main effects of job-related efforts, rewards, and over-commitment on WLC were found, while perceived schedule flexibility and work–life integration were also found to significantly reduce WLC [33]. In turn, it has been found that a poor work–life balance is associated with poor psychosocial well-being [35].

This study analyses gender differences in the indirect effects of a set of psychosocial risk factors (i.e., psychological demands, control, and social support) and work–life conflict on the relationship between multidimensional PE and the production of steroid hormones (both adrenal and gonadal) in salaried workers from Barcelona city.

## 2. Materials and Methods

### 2.1. Study Design and Sampling

A cross-sectional study was conducted based on a sample of 255 employees from Barcelona, Spain, aged 25–60 years (125 men and 130 women), as described elsewhere [6]. Briefly, the sample design was non-probabilistic and based on proportional quotas determined by sex, age group, place of birth, and socioeconomic level of the district of residence. Participants were recruited from a pool of participants from the 2017 Barcelona Health Survey within the selected age range who had agreed to being contacted again for future studies and agreed specifically to being contacted by the University for this project (n = 1210). Furthermore, the abovementioned recruitment strategy was complemented with 40 individuals contacted through social and labour organizations in order to offset the bias of this subsample toward profiles with higher levels of education and income. Under the assumptions of an alpha risk of 0.05 and a beta risk below 0.2 (80% power) in a two-sided test, and with a sample loss rate of 0%, our sample size allowed us to estimate correlation coefficients of 0.175. Without exemptions, the inclusion criteria were: (i) being a salaried worker or an independent worker, (ii) being between 24 and 60 years old, (iii) living independently in Barcelona, (iv) having hair at least 1 cm in length on the back of the head, and (v) not having been off work (on holiday or on work leave) within the month prior to the interview. Exclusion criteria were: (i) having taken corticosteroids in the month prior to the interview, (ii) having an adrenal disease, and (iii) being pregnant, as gestation could alter cortisol levels. 

A face-to-face interview of approximately 40 min was conducted with each sample subject in which a questionnaire was administered including questions on the topics of interest for the study (PE, working conditions, uncertainty, support networks, perceived stress, and physical and mental health), as well as items on sociodemographic characteristics.

The first centimetre of the lock of hair in contact with the scalp was sent for analysis in order to obtain information on steroids produced during the month prior to sample collection. 

### 2.2. Variables

Outcome variables. A comprehensive steroid profile was measured from hair using a previously validated method based on liquid chromatography–tandem mass spectrometry [36]. Steroids and metabolites were divided between adrenal steroids (providing information about the HPA axis)—including 20α-dihydrocortisol (20αDHF), 20ß-dihydrocortisol (20βDHF), cortisone, 20α-dihydrocortisone (20αDHE), 20ß dihydrocortisone (20βDHE), cortolone, 11-dehydrocorticosterone, and androstenedione hair cortisol levels—and gonadal steroids (providing information about the HPG axis)—including androstenedione, testosterone, and progesterone levels. In addition to the hair concentrations of the targeted steroids, several ratios were included in order to evaluate the activity of key enzymes in the production and metabolism of steroids. For example, the cortisol/cortisone ratio was calculated to evaluate the activity of the enzyme 11ß-hydroxysteroid dehydrogenase (responsible for the interconversion between cortisol and cortisone). Additionally, the relative abundance of each glucocorticosteroid (in percent) was calculated as an additional marker. Since the distributions of the steroids and ratios were very dissimilar, the natural logarithm was used to fit them to a normal distribution and obtain more reliable statistics. 

Explanatory variables. PE was measured through an adaptation of the EPRES scale validated with the PRESSED data. The psychometric properties of this scale have been published elsewhere [6]. The scale consists of 24 indicators sorted into the dimensions of the EPRES scale specified above and another dimension related to extra working hours. Each dimension contributed equally to the total score, regardless of its number of items. To obtain an equal-weight scale, each dimension score was computed independently, standardized, and integrated into a global summary score. Accordingly, the items in each dimension were added together, and the overall score was transformed into a 0 to 4 score. Then, these scores were averaged into a global EPRES score, which ranged from 0 (not precarious) to 4 (most precarious) [21]. 

Mediators. The WLC and PRF dimensions “Psychological Demands”, “Control”, and “Social Support” were measured using 32 items from the COPSOQ III [37]. Scores for each dimension were computed through simple averages of its corresponding items. Exploratory analyses, confirmatory factor analyses, and Cronbach’s alpha coefficients were used to evaluate the scales’ validity and reliability, respectively (see Appendix A). Regarding the validity, factor-loading estimates revealed that all items were related to their theorized dimensions. The scale exhibited acceptable psychometric properties and reliability (ω  =  0.78 for the “Demands” score; ω  =  0.71 for the “Control” score; ω  =  0.76 for the “Support” score; ω  =  0.81 for the “Work–Life Conflict” score). The factor structure was confirmed with CFA (χ^2^ (df)  =  732.672 (203), *p*  <  0.0001; CFI  =  0.939; TLI  =  0.930; RMSEA (95% CI)  =  0.104 (0.096–0.112); all paths statistically significant).

Covariates. The covariates used for adjustment were age (continuous), body mass index (BMI), occupational social class (i.e., “Manual”, “Non-manual”), and a proxy of care work (people younger than 14 years old at home).

### 2.3. Statistical Analysis

A description of the studied sample was produced. Means and their standard deviation were calculated for continuous variables and prevalence and 95% CIs for categorical variables.

Linear regression models were fitted to estimate the association between PE, PRFs, and WLC and steroid production. Two models were estimated. Model 1 (crude) was adjusted for age and BMI. Model 2 (adjusted) was further adjusted for occupational social class, care work, demands, control, social support, and WLC. 

The Karlson–Holm–Breen (KHB) method was used to estimate the indirect effects of PRFs and WLC on the relationship between PE and markers. Two models were fitted. Model 1 included demands, control, and social support as mediator variables, while in model 2, WLC was added as a mediator. Both models were adjusted for age, BMI, occupational social class, and care work. The KHB method allows the unbiased comparison of regression coefficients between models and the decomposition of mediation effects [38]. This method, therefore, allowed us to compare a reduced model with a full model. In the reduced model, PE predicted markers from two approaches: one fitting crude models, the other controlling for covariates. In the full model, PRFs were introduced as “Z variables”, and the model was controlled for them. For theoretical reasons, these Z variables were conceptualized as mediators that influence markers. The difference in coefficients between the reduced and full models represented the spurious component of the effect of PE on markers, which included the association mediated by PRFs and WLC. Standard errors were adjusted for heteroscedasticity to obtain a robust estimate of variance. All the analyses were stratified by sex and conducted using Stata 16.0. Results were considered significant when *p* < 0.05. 

### 2.4. Ethics

This study was reviewed and approved by the Institutional Commission for Ethics Review of Projects (CIREP) of Universitat Pompeu Fabra (UPF) under CIREP number 0079. All participants in the study signed a written informed consent form prior to the start of the study and were reminded that they could withdraw at any time. All data used in this project followed the laws of Spain governing personal data protection, fulfilling all legal and ethical requirements. They were duly processed, preserving data anonymity and confidentiality.

## 3. Results

### 3.1. Descriptive Results

The characteristics of the sample studied are shown in Table 1. Gender differences in markers levels were significant for 20αDHF, 20αDHE, cortisone, androstenedione, cortolone, testosterone, %20αDHE, and 20αDHF/cortisol, which were higher among men; and cortisol/cortisone, %cortisol, %20αDHF, %20βDHF, %20βDHE, and 20αDHF/20βDHF, which were higher among women. No gender differences were found for PE. Regarding PRFs, “Demands” was higher among women (0.47; 95% CI: 0.44–0.50 vs. 0.41; 95% CI: 0.38–0.45). Gender differences for WLC were not found. 

### 3.2. Precarious Employment, Psychosocial Risk Factors, and Production of Adrenal and Gonadal Steroids

The association between markers and PE and PRFs concomitantly is presented in Table 2. Linear regression coefficients adjusted for control variables are shown in Table 2a for men and Table 2b for women. Among men, androstenedione (β = 0.22; 95% CI: 0.01–0.44) and testosterone (β = 0.26; 95% CI: 0.02–0.50) were associated with PE, after adjusting for PRFs. Concerning PRFs, 20αDHF/20βDHF (β = 0.68; 95% CI: 0.01–1.34), cortisol/cortisone (β = 0.66; 95% CI: 0.13–1.20), and %cortisol (β = 0.53; 95% CI: 0.16–0.90) were positively associated with “Demands”, whilst 20αDHF/cortisol (β = −0.66; 95% CI: −1.15–−0.18) was negatively associated with this PRF. There were no steroids associated with “Control”. “Social Support” showed positive coefficients for %cortisone (β = 0.28; 95% CI: 0.04–0.51) and negative coefficients for 20αDHF (β = −0.95; 95% CI: −1.84–−0.07), 20βDHF (β = −0.71; 95% CI: −1.35–−0.07), and cortisol/cortisone (β = −0.57; 95% CI: −1.06–−0.08). WLC was negatively associated with 20αDHF/20βDHF (β = −0.57; 95% CI: −1.09–−0.06). 

Among women, negative associations between PE and cortisol/cortisone (β = −0.35; 95% CI: −0.56–−0.14) and %cortisol (β = −0.24; 95% CI: −0.38–−0.09) were found after adjusting for PRFs. Concerning PRFs, “Demands” showed positive coefficients for cortisol/cortisone (β = 1.02; 95% CI: 0.50–1.54), %cortisol (β = 0.59; 95% CI: 0.21–0.97), %20αDHF (β = 0.66; 95% CI: 0.18–1.14), and %20αDHE (β = 0.37; 95% CI: 0.12–0.61) and negative coefficients for androstenedione (β = −0.95; 95% CI: −1.63–−0.27), cortisone/dehydrocorticosterone (β = −0.78; 95% CI: −1.41–−0.15), and %cortisone (β = −0.47; 95% CI: −0.66–−0.27). “Control” was positively associated with cortisol/cortisone (β = 0.94; 95% CI: 0.35–1.53) and %cortisol (β = 0.52; 95% CI: 0.11–0.93) and negatively associated with %cortisone (β = −0.39; 95% CI: −0.64–−0.14). Cortisone (β = 0.49; 95% CI: 0.11–0.86), cortisone/dehydrocorticosterone (β = 0.64; 95% CI: 0.14–1.13), and %cortisone (β = 0.37; 95% CI: 0.19–0.55) were positively associated with “Social Support”, whilst 20αDHE/20βDHE (β = −0.25; 95% CI: −0.45–−0.04), cortisol/cortisone (β = −0.53; 95% CI: −1.00–−0.06), %20αDHF (β = −0.52; 95% CI: −0.86–−0.18), %20βDHF (β = −0.20; 95% CI: −0.40–−0.00), %20αDHE (β = −0.45; 95% CI: −0.66–−0.24), and %20βDHE (β = −0.20; 95% CI: −0.38–−0.03) were negatively associated with this PRF. WLC showed a negative coefficient for dehydrocorticosterone (β = −0.41; 95% CI: −0.78–−0.03).

### 3.3. Precarious Employment and Production of Adrenal and Gonadal Steroids: The Indirect Effect of Psychosocial Risk Factors

Table 3 shows the results of KHB decomposition analyses for those steroids that were associated with PE. Thus, the indirect effect of PE through the PRFs and work–life conflict could be estimated, while the comparison between model 1 and model 2 made it possible to estimate the change in the indirect effect when work–life conflict was added as a mediating variable.

Among men, no indirect effect of PRFs was observed. There were no indirect effects when adding WLC as a mediator, although a significant total effect was observed for both gonadal steroids ((β_Androstenedione_: 0.22; 95% CI: 0.02–0.43)); (β_Testosterone_: 0.26; 95% CI: 0.02–0.50)). Among women, indirect effects for some adrenal steroids, such as cortisol (β = 0.18; 95% CI: 0.04–0.32), cortisol/cortisone (β 0.19; 95% CI: 0.08–0.31), and %cortisol (β 0.12; 95% CI: 0.05–0.20), were found in model 1. When incorporating WLC as a mediator, there was no indirect effect for cortisol, and the magnitude of the effect decreased for the other steroids (β_Cortisol/Cortisone_: 0.15; 95% CI: 0.02–0.28; β_% Cortisol_: 0.09; 95% CI: 0.01–0.18).

## 4. Discussion

Following the results of a previous PRESSED-related article where associations between PE and steroid hormones were found [6], the main objective of this study was to explore the role of the psychosocial work environment as a possible mediator in such a relationship. To achieve this objective, the statistical analysis was performed in two steps: firstly, the association between PE and steroid hormone production was estimated, adjusting for PRF and WLC, as well as for covariates. Secondly, the indirect effects of PRFs and WLC on this relationship were estimated.

### 4.1. The Association between Precarious Employment and Steroid Hormone Production, Adjusting for PRF and WLC

The main results showed that the association between PE and steroid hormone production was maintained when adjusting for PRFs and WLC, with remarkable differences between men and women. A positive association between androgens—i.e., gonadal steroids, (AED and testosterone)—and PE was found among men. In contrast, women showed a negative association between PE and corticosteroids; i.e., adrenal steroids (cortisol and metabolites). Several potential explanations might lie behind these results. 

From a biochemical point of view, gender differences in the production of steroid hormones and their relationship with stress have been previously described [39]. The results of this study suggested that PE would increase the production of gonadal steroids among men, leading to a subsequent rise in aggressiveness and dominant behaviour [40], thus highlighting the pivotal role of the HPG axis in men. In contrast, it was found that the role of the HPA axis is more important for women. The key function of the HPA axis in the relationship between PE and steroids is not surprising since overproduction of cortisol is a common biological feature under stress conditions. More surprising is the negative association observed between PE and several metabolites related to the HPA axis. Although several studies have shown gender differences in cortisol production after stressful events [41], it is difficult to conclude that the negative correlation is exclusively due to biochemical reasons.

Sociological factors, such as working conditions, might also exacerbate gender differences in response to stressful situations. Previous studies have found gender differences in occupational health related to working conditions (Campos-Serna et al., 2013) that were linked to structural gender inequality in labour markets (Menéndez et al., 2007). A preliminary hypothesis could be that the physiological effect of PE is influenced by the social construction of gender identities, differentially affecting men and women. In a patriarchal system marked by the sexual division of labour, in which the masculine role mainly draws on the “male breadwinner” stereotype, men can be psychologically affected by the perception of not meeting the social expectations associated with their role. Thus, the increased production of gonadal steroids could be a way for men to respond psychophysiologically to PE. This hypothesis is based on classic social psychology approaches suggesting that the impact of employment problems on health is related to the different positions and roles available for men and women in society and the family [42]. For example, it has been found that unemployment was more negatively related to mental health among men than among women in a gender regime in which the need for employment differed between sexes (Ireland), while men and women were equally affected by unemployment in a gender regime where there was a similar need for employment (Sweden) [43].

Therefore, men and women have different psychosocial and economic employment needs based on gender roles [44]. In fact, in this study, we found that, among women, the association between some steroids and PE presented negative coefficients, showing an inverse relationship to that hypothesised. This may have been because, unlike men, women’s perceptions of PE are not influenced by the role of providers. Furthermore, the position of women in the sexual division of labour as the main partner responsible for care and home duties may imply that some characteristics of PE, such as flexibility or a low workload, are perceived as beneficial because they contribute to reconciling paid and unpaid work [45].

It should be noted that, although the sexual division of labour has been losing its rigidity over time, mainly due to the massive and sustained entry of women into the job market (Soares and Falcão, 2015), there has not been any effective redistribution of responsibilities within the family, where changes are slower and co-responsibility between men and women is still a long way off [46,47]. Furthermore, gender relations within family frameworks still tend to be patriarchal, and even if occupational status is higher, women rarely have enough power to force men to agree to an equitable division of domestic work and childcare [48,49,50].

### 4.2. The Indirect Effect of Psychosocial Risk and Work–Life Conflict in the Association between Precarious Employment and Steroid Hormones Production

Regarding the psychosocial work environment, it was found that, for both men and women, high demands and low social support were the two psychosocial factors associated with the production of the highest number of steroids. Although the meaning of these associations is not entirely conclusive, it is noteworthy that, for low social support, the associations with most steroids were negative, while for high demands, the majority were positive. This implies that, while low social support increases steroid production, high demands reduce it, suggesting that the latter could be a protective mechanism. In this sense, several previous studies of mental health have found that high demands reduce the risk of depression and anxiety disorders [51,52].

At the same time, the existence of indirect effects of PRFs and WLC on the relationship between PE and steroids production would indicate that a proportion of the association between the exposure and the outcome of interest crystallizes through the psychosocial work environment. The results show significant indirect effects only for women, suggesting the existence of gender differences in the psychophysiological response to PE. A recent study found a full mediation by PRFs of PE and mental health among women and a partial mediation among men. The authors suggest that women are more exposed to worse working conditions, including the psychosocial environment, due to occupational segregation of gender in the labour market [53]. Both the findings show that the psychosocial work environment has a greater weight in women’s psychological and physiological responses to PE than with men. Thus, women may react more to proximal factors, such as the psychosocial environment, than to distal factors, such as PE, while precisely the opposite occurs among men. The study does not allow further progress in determining the possible causes of these differences. However, it will be necessary to delve into gender differences in perceptions of working and employment conditions in future studies.

### 4.3. Strength and Limitations

This study has some limitations inherent to a cross-sectional design. Firstly, it was not possible to estimate a direct causal effect, and possible reverse causality must be considered: high production of steroids (which could indicate psychophysiological alterations) at the beginning of the study may have increased the chances of having a precarious job or a hazardous psychosocial environment. Second, there was no information on the period during which these individuals were exposed to PE or PRFs, which may have somewhat altered the results. Therefore, further longitudinal studies are needed.

On the other hand, a notable strength of this article was its use of biological markers, something new not just in the study of PE but also in the field of social epidemiology, where subjective and/or self-reported health measures are usually used. In turn, in biochemical research, simultaneously studying the two axes (gonadal and adrenal) in hair is also new. In previous studies, only steroids of the adrenal axis have been studied.

At the same time, this article shows the importance of considering employment conditions in the study of psychosocial working conditions. Most psychosocial risk models theoretically assume social causality, where the organization of work determines the psychosocial work environment, but they do not explain the individual’s relationship with the environment [54]. Furthermore, assimilating the “social” to the “psychological” means that the models are unable to explain how the social structure determines the psychosocial work environment [22]. Taking PE into account allows us to explain how the political context and labour relations determine the organization of work in a complex process that impacts on workers’ health. Therefore, empirical advances such as those offered in this article stimulate the development of new theoretical and methodological frameworks that relate psychosocial risks to PE to explain the global impact of the workplace on health.

## 5. Conclusions

Gender differences were found in the association between PE and the production of steroid hormones (both adrenal and gonadal) and in the indirect effects of PRFs and WLC. Biochemically, this could indicate the pivotal role of the HPG axis in men, the HPA axis being more important for women. In turn, these results suggest that the physiological effect of PE could be mediated by the social construction of gender identities, which draws on the “male breadwinner” stereotype. This contributes to supporting the hypothesis that the influence of PE on health is related to the different positions and roles of men and women in society and the family. Future studies should delve further into these differences in the relationships between PE, PRFs, and their psychophysiological effects to improve employment and working policies, especially from the perspective of the social determinants of health. However, this hypothesis should be evaluated based on research designs and instruments that make it possible to understand the daily dynamics of work organization and how the psychosocial factors that put the health of female workers at risk are produced. Thus, it is necessary to conduct qualitative studies that make it possible to capture the experiences and perceptions of men and women within their daily work context. 

Furthermore, this study highlights the importance of relating the psychosocial work environment to employment conditions in order to explain how structural factors (such as labour relations and employment policies) influence working conditions and affect health. Nevertheless, there are even more structural processes that precede PE and configure it, such as the fragmentation of workers’ representation, holding multiple jobs, and the complexity of companies and organizations in the global economy. More studies are needed to analyse the impact of these phenomena on organizational structures and the employment and working conditions generated within them.

## Figures and Tables

**Table 1 ijerph-19-16073-t001:** Characteristics of the study population stratified by sex. Precarious Employment and Stress Study sample, 2020 (95% CI = 95% confidence interval, SD = standard deviation).

	Men (*n* = 125)		Women (*n* = 130)		*p*-Value
	Proportion	95% CI	Proportion	95% CI	
Occupational social class					
No manual	71.20	(39.95–43.41)	76.15	(68.76–83.54)	.
Manual	28.80	(20.79–36.81)	23.85	(16.46–31.24)	0.820
	**Mean**	**SD ^1^**	**Mean**	**SD ^1^**	
Age (years)	41.68	9.84	42.75	9.79	0.383
Body mass index (BMI) (kg/m^2^)	25.34	3.59	24.75	4.31	0.233
Care work	0.54	0.77	0.61	0.85	0.480
Precarious employment (EPRES)	1.04	0.56	1.02	0.55	0.797
Psychosocial risk factors					
Demands	0.41	0.18	0.47	0.19	0.015
Control	0.30	0.18	0.82	0.17	0.440
Support	0.29	0.20	0.28	0.21	0.800
Leadership	0.43	0.21	0.39	0.20	0.176
Work–life conflict	0.44	0.22	0.47	0.26	0.299
Adrenal and gonadal steroids (ng/mg)					
Cortisol	12.42	18.90	9.33	7.35	0.090
20αDHF	0.99	1.15	0.99	0.96	0.977
20βDHF	5.67	4.19	4.46	2.71	0.007
20αDHE	10.20	8.42	9.05	6.41	0.223
20βDHE	7.48	5.70	5.37	3.26	0.000
Cortisone	33.89	18.37	26.88	17.20	0.002
Cortolone	8.73	3.94	7.09	3.08	0.000
11-Dehidrocorticosterone (A)	2.99	1.59	2.54	1.57	0.023
Testosterone	2.07	2.10	3.08	24.36	0.638
Androstenedione (AED)	5.43	3.07	3.91	2.90	0.000
Progesterone	232.62	1511.24	27.01	32.96	0.130
20αDHF/20βDHF	0.17	0.17	0.20	0.10	0.048
20αDHE/20βDHE	1.38	0.33	1.65	0.39	0.000
Cortisone/11-dehydrocorticosterone (E/A)	13.01	7.09	12.97	8.82	0.968
Cortisol/cortisone	0.36	0.45	0.36	0.24	0.901
%Cortisol	15.12	9.25	16.04	6.42	0.359
%Cortisone	50.54	10.29	48.51	9.16	0.097
%20αDHF	1.23	0.76	1.61	0.82	0.000
%20βDHF	8.18	2.74	8.24	2.15	0.843
%20αDHE	14.30	3.92	15.85	4.39	0.003
%20βDHE	10.63	2.80	9.75	2.18	0.006
20αDHF/cortisol	0.09	0.06	0.11	0.08	0.030
20βDHF/cortisol	0.65	0.31	0.59	0.28	0.074

^1.^ Standard Deviation.

**Table 2 ijerph-19-16073-t002:** Linear regression coefficients and 95% confidence intervals (CIs) for production of adrenal and gonadal steroids and PE and PRFs concomitantly, adjusted for control variables and stratified by sex. Precarious Employment and Stress Study sample, 2020.

(a)—Men	EPRES		Psychosocial Risk Factors						Work−Life Conflict	
Adrenal and Gonadal Steroids			Demands		Control		Support			
20αDHF	0.23	(−0.14–0.59)	0.53	(−0.53–1.59)	0.94	(−0.26–2.13)	−0.95 *	(−1.84–−0.07)	−0.55	(−1.40–0.30)
20βDHF	0.09	(−0.15–0.33)	−0.03	(−0.73–0.67)	0.57	(−0.24–1.39)	−0.71 *	(−1.35–−0.07)	−0.07	(−0.61–0.47)
Androstenedione	0.22 *	(0.01–0.44)	−0.49	(−1.10–0.12)	−0.22	(−0.82–0.38)	−0.13	(−0.76–0.50)	−0.30	(−0.82–0.23)
Testosterone	0.26 *	(0.02–0.50)	−0.33	(−1.12–0.46)	−0.24	(−1.00–0.53)	−0.47	(−1.19–0.24)	−0.18	(−0.74–0.38)
20αDHF/20βDHF	0.12	(−0.05–0.29)	0.68 *	(0.01–1.34)	0.08	(−0.67–0.83)	0.05	(−0.60–0.70)	−0.57 *	(−1.09–−0.06)
Cortisol/cortisone	−0.08	(−0.28–0.13)	0.66 *	(0.13–1.20)	0.30	(−0.28–0.88)	−0.57 *	(−1.06–−0.08)	0.16	(−0.28–0.60)
%Cortisol	−0.08	(−0.24–0.08)	0.53 **	(0.16–0.90)	0.18	(−0.23–0.58)	−0.25	(−0.60–0.09)	0.10	(−0.22–0.42)
%Cortisone	0.00	(−0.08–0.08)	−0.11	(−0.38–0.17)	−0.12	(−0.42–0.19)	0.28 *	(0.04–0.51)	−0.09	(−0.29–0.12)
20αDHF/cortisol	0.11	(−0.08–0.30)	−0.66 **	(−1.15–−0.18)	−0.09	(−0.63–0.46)	0.02	(−0.44–0.48)	0.01	(−0.38–0.41)
**(b)—Women**	**EPRES**		**Psychosocial Risk Factors**						**Work−Life Conflict**	
**Adrenal and Gonadal Steroids**			**Demands**		**Control**		**Support**			
Cortisone	0.09	(−0.11–0.29)	−0.47	(−1.04–0.10)	−0.33	(−1.00–0.34)	0.49 *	(0.11–0.86)	−0.02	(−0.47–0.43)
Androstenedione	0.20	(−0.07–0.47)	−0.95 **	(−1.63–−0.27)	−0.15	(−1.04–0.74)	−0.06	(−0.61–0.48)	0.13	(−0.35–0.61)
Dehydrocorticosterone	0.12	(−0.14–0.37)	0.26	(−0.22–0.74)	0.38	(−0.25–1.01)	−0.16	(−0.52–0.20)	−0.41 *	(−0.78–−0.03)
20αDHE/20βDHE	−0.03	(−0.13–0.06)	0.16	(−0.09–0.40)	0.11	(−0.17–0.38)	−0.25 *	(−0.45–−0.04)	−0.05	(−0.22–0.12)
Cortisone/dehydrocorticosterone	−0.01	(−0.32–0.30)	−0.78 *	(−1.41–−0.15)	−0.52	(−1.29–0.26)	0.64 *	(0.14–1.13)	0.38	(−0.17–0.94)
Cortisol/cortisone	−0.35 **	(−0.56–−0.14)	1.02 **	(0.50–1.54)	0.94 **	(0.35–1.53)	−0.53 *	(−1.00–−0.06)	−0.28	(−0.69–0.13)
%Cortisol	−0.24 **	(−0.38–−0.09)	0.59 **	(0.21–0.97)	0.52 *	(0.11–0.93)	−0.16	(−0.49–0.17)	−0.21	(−0.50–0.08)
%Cortisone	0.09	(−0.00–0.17)	−0.47 **	(−0.66–−0.27)	−0.39 **	(−0.64–−0.14)	0.37 **	(0.19–0.55)	0.10	(−0.07–0.27)
%20αDHF	−0.09	(−0.30–0.12)	0.66 **	(0.18–1.14)	0.56	(−0.00–1.13)	−0.52 **	(−0.86–−0.18)	−0.08	(−0.45–0.30)
%20βDHF	−0.06	(−0.17–0.05)	0.26	(−0.01–0.53)	0.29	(−0.06–0.64)	−0.20 *	(−0.40–−0.00)	0.02	(−0.19–0.23)
%20αDHE	0.01	(−0.11–0.13)	0.37 **	(0.12–0.61)	0.15	(−0.14–0.43)	−0.45 **	(−0.66–−0.24)	−0.04	(−0.24–0.16)
%20βDHE	0.04	(−0.05–0.13)	0.20 *	(−0.03–0.44)	0.04	(−0.19–0.28)	−0.20 *	(−0.38–−0.03)	0.01	(−0.17–0.19)

** *p* < 0.01, * *p* < 0.05.

**Table 3 ijerph-19-16073-t003:** Linear regression coefficients and 95% confidence intervals (CIs) for the production of adrenal and gonadal steroids and PE and PRFs, adjusted for control variables and stratified by sex, using the KHB method. Robust standard errors. Precarious Employment and Stress Study sample, 2020.

(a)—Men	Direct		Total		Indirect	
Model 1	Coeff.	95% CI	Coeff.	95% CI	Coeff.	95% CI
Androstenedione (AED)	0.13	(−0.08–0.35)	0.17	(−0.04–0.39)	−0.04	(−0.10–0.02)
Testosterone	0.18	(−0.05–0.40)	0.23	(−0.01–0.46)	−0.05	(−0.13–0.02)
Model 2	Coeff.	95%CI	Coeff.	95%CI	Coeff.	95% CI
Androstenedione (AED)	0.13	(−0.07–0.34)	0.22 *	(0.00–0.44)	−0.09	(−0.20–0.02)
Testosterone	0.18	(−0.05–0.40)	0.26 *	(0.02–0.49)	−0.08	(−0.20–0.04)
**(b)—Women**	**Direct**		**Total**		**Indirect**	
Model 1	Coeff.	95% CI	Coeff.	95% CI	Coeff.	95% CI
Cortisol/cortisone	−0.23 *	(−0.40–−0.05)	−0.39 **	(−0.59–−0.20)	0.17 **	(0.06–0.27)
%Cortisol	−0.17 **	(−0.30–−0.04)	−0.27 **	(−0.41–−0.14)	0.10 **	(0.03–0.17)
Model 2	Coeff.	95% CI	Coeff.	95% CI	Coeff.	95% CI
Cortisol/cortisone	−0.23 **	(−0.40–−0.06)	−0.35 **	(−0.56–−0.15)	0.12 *	(0.00–0.24)
%Cortisol	−0.17 **	(−0.29–−0.05)	−0.24 **	(−0.38–−0.10)	0.07	(−0.01–0.15)

** *p* < 0.01, * *p* < 0.05.

## Data Availability

The data presented in this study are available on request from the corresponding author. The data are not publicly available due to their containing information that could compromise the privacy of research participants.

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
