# Peer review of "Gender Differences in the Indirect Effect of Psychosocial Work Environment in the Association of Precarious Employment and Chronic Stress: A Cross-Sectional Mediation Analysis"

_ijerph, 2022, doi:10.3390/ijerph192316073_

Round 1

Reviewer 1 Report

Dear authors,

It is an honor for me to review your manuscript entitled “Gender differences in the indirect effect of psychosocial work environment in the association of precarious employment and chronic stress: A cross-sectional mediation analysis”.

The topic it addresses is of great importance and relevance both in the area of ​​employment and in the area of ​​occupational health.

In this regard, I have several proposals for improvement:

The introduction has some studies from the last 5 years, but in general it is somewhat obsolete and little worked on. On the one hand, when dealing with precarious jobs or precarious working conditions and stress, it would be advisable to include a paragraph where these precarious situations are analyzed: little intrinsic or extrinsic motivation, lack of social skills, lack of leadership skills in superiors, etc. On the other hand, I recommend that you include some studies from the last two years to add novelty and reduce the rate of obsolescence of the analysis of the state of the art. I recommend, among others:

Abu-Kaf, S., y Khalaf, E. (2020). Acculturative Stress among Arab Students in Israel: The Roles of Sense of Coherence and Coping Strategies. International Journal of Environmental Research and Public Health, 17(14), 5106. https://doi.org/10.3390/ijerph17145106

Agarwal, A., Ali, H., Sa, S., y Hussain Shah, J. (2020). Resilience and Burnout: Relation of Emotional Intelligence (EI) and Stress Management Capabilities among Health Professional Students of Northern Border University -Arar, Kingdom of Saudi Arabia. PAKISTAN JOURNAL OF MEDICAL y HEALTH SCIENCES, 14(1), 425–429.

Sánchez-Bolívar, L., Escalante-González, S., & Martínez-Martínez, A. (2022). Motivation and Social Skills in Nursing Students Compared to Physical Education Students . SPORT TK-EuroAmerican Journal of Sport Sciences, 11, 5. https://doi.org/10.6018/sportk.462121

Aslan, I., Ochnik, D., y Çınar, O. (2020). Exploring Perceived Stress among Students in Turkey during the COVID-19 Pandemic. International Journal of Environmental Research and Public Health, 17(23), 8961. https://doi.org/10.3390/ijerph17238961

Farr, J., Ononaiye, M., y Irons, C. (2021). Early shaming experiences and psychological distress: The role of experiential avoidance and self‐compassion. Psychology and Psychotherapy: Theory, Research and Practice. https://doi.org/10.1111/papt.12353

Ma, Y., Peng, H., Liu, H., Gu, R., Peng, X., y Wu, J. (2021). Alpha frontal asymmetry underlies individual differences in reactivity to acute psychosocial stress in males. Psychophysiology. https://doi.org/10.1111/psyp.13893

Regarding the method, the assumed sampling error is not reflected in the sample. Must be included.

The psychometric properties of the instruments should be delved into: types of scale, dimensions it measures, original validity and reliability, and the global reliability obtained in the study.

A sub-heading “Procedure” must be included, reflecting what procedure was followed, how consent was requested, what code of ethics was followed in the research and the phases of the research process.

In the results, the tables must be framed and reduced in size, specifically table 2 occupies two pages. Reduce it to one page. Pages 9 and 10 have to be unified.

In the Conclusions, several paragraphs must be included with the limitations that have arisen during the investigation, as well as the future perspectives that have arisen as a consequence of this study.

Reviewer 2 Report

This is a very interesting and important study connecting "embodiment" research on chronic psychosocial stress/physiological markers and gender in a context of precarious employment (PE). The authors found male workers who experience PE had higher androgen levels, whereas for women, there was a negative association. Women with PE had lower levels of corticosteroids (markers of stress), a surprising finding. The authors suggest this might be partly explained by the gender division of labor. Men with precarity might feel their gender identities as "breadwinners" are under threat and produce more male hormones (e.g. testosterone) in response. For women, whose gendered division of labor unfortunately still involves much more responsibility for the domestic and child care work, may find some flexibility in precarious work so they can better balance work and non-work life. The authors rightly point out that further research is needed to confirm these hypotheses.

The findings about psychosocial risk factors (PRF) were surprising regarding high job demands being related to lower corticosteroid levels. It is possible these findings may be related to higher SES or individuals who are in the "active" quadrant of the job strain model. If they have high job demands and high job control this is considered "health protective". More needs to be said about this.

The finding that women may "react" more to proximal psychosocial factors in the workplace by producing more corticosteroids whereas men in this sample with PE had higher levels of stress hormones, was really interesting and tends to align with the literature. I agree with the authors that this may be due to occupational sex segregation where women's work has a greater exposure in general to PRF.

The only suggestion I have is to add a statement regarding human subjects consent and institutional review of this research. 

I think this research article is very deserving of publication and will contribute to the important work being done in biopsychosocial studies of chronic workplace stress.

Round 2

Reviewer 1 Report

Dear authors,

Dear authors,

I pointed out that their manuscript was generally out of date and that they should include references to the last 5 years in the introduction and discussion. This they did not. They limited themselves to using 4 references, which they already had, in a new paragraph.

On the one hand, when dealing with precarious jobs or precarious working conditions and stress, it would be advisable to include a paragraph where these precarious situations are analyzed: little intrinsic or extrinsic motivation, lack of social skills, lack of leadership skills in superiors, etc. On the other hand, I recommend that you include some studies from the last two years to add novelty and reduce the rate of obsolescence of the analysis of the state of the art. For this, I recommended some references, in order to facilitate their work. They did not use them, nor did they delve into what I was exposing to them.

Likewise, I told them that, to facilitate reading, they should include the Procedure followed in the investigation and the ethical code respected in it. In their response, they limit themselves to defending that they locate it in another place, ignoring my advice.

Finally, I indicate that they have to extract the limitations of the discussion and reflect them at the end of the conclusions. Once again, they are justified in not applying this correction. Based on the foregoing, I indicate that, in the discussion, the results are discussed with studies for and against. The limitations of an investigation are elucidated when its conclusions are drawn, which is why they are located in the Conclusions section, together with the future research perspectives that we will use to cover such limitations.

After a second revision, I am sorry to see that they have not applied all of my modifications. As suggested in the review, the manuscript required profound modifications for its publication and you have limited yourself to applying superficial modifications.
